# Are Microplastics Impairing Marine Fish Larviculture?—Preliminary Results with *Argyrosomus regius*

Diana Campos [1], Andreia C. M. Rodrigues [1,*], Rui J. M. Rocha [1], Roberto Martins [1], Ana Candeias-Mendes [2], Sara Castanho [2], Florbela Soares [2], Pedro Pousão-Ferreira [2], Amadeu M. V. M. Soares [1], Carlos Gravato [3] and Ana L. Patrício Silva [1]

1    CESAM—Centre for Environmental and Marine Studies, Department of Biology, University of Aveiro, 3810-193 Aveiro, Portugal; diana.campos@ua.pt (D.C.); ruimirandarocha@ua.pt (R.J.M.R.); roberto@ua.pt (R.M.); asoares@ua.pt (A.M.V.M.S.); ana.luisa.silva@ua.pt (A.L.P.S.)
2    IPMA—Portuguese Institute for the Ocean and Atmosphere, EPPO—Aquaculture Research Station, Av. Parque Natural da Ria Formosa s/n, 8700-194 Olhão, Portugal; ana.mendes@ipma.pt (A.C.-M.); scastanho@ipma.pt (S.C.); fsoares@ipma.pt (F.S.); pedro.pousao@ipma.pt (P.P.-F.)
3    Faculty of Sciences and CESAM, University of Lisbon, Campos Grande, 1749-016 Lisbon, Portugal; cagravato@fc.ul.pt
*    Correspondence: rodrigues.a@ua.pt

**Abstract:** The presence of small-sized (<300 μm) microplastics (MPs) in aquaculture facilities may threaten finfish hatchery, as their (in)voluntary ingestion by fish larvae may compromise nutritional requirements during early ontogeny, and consequently larval health and performance. Thus, we addressed the short-term effects (7 h) of polyethylene microplastics (0.1, 1.0, 10 mg/L, PE-MPs) in meagre larvae *Argyrosomus regius* (15 dph) in the presence/absence of food. Larval feeding behavior, oxidative stress status, neurotoxicity, and metabolic requirements were evaluated. Results showed that meagre larvae ingested PE-MPs regardless of their concentration, decreasing in the presence of food (*Artemia metanauplii*). The presence of PE-MPs compromised larval feeding activity at the highest concentration. Under starvation, exposed larvae activated the antioxidant defenses by increasing the total glutathione levels and inhibiting catalase activity, which seemed efficient to prevent oxidative damage. Such larvae also presented increased energy consumption potentially related to oxidative damage prevention and decreased neurotransmission. Biochemical responses of fed larvae showed a similar trend, except for LPO, which remained unaffected, except at 0.1 mg/PE-MPs/L. Our results suggest that small-sized MPs in finfish hatcheries may compromise larvae nutritional requirements, but at considerably higher levels than those reported in marine environments. Nevertheless, cumulative adverse effects due to lower MPs concentrations may occur.

**Keywords:** plastic pollution; polyethylene; oxidative stress status; neurotoxicity; farmed fish species





## 1. Introduction

Aquaculture represents an essential food source for the world's human population, contributing significantly to welfare and food security. For this reason, aquaculture is being prioritized by the 2030 Agenda, and it is expected to reach 109 million tonnes by 2030 [1]. The success and sustainability of aquaculture at a global scale rely on hatcheries, where breeding, egg fertilization, incubation, hatching, and rearing of organisms, particularly finfish, through early-life stages is carried out artificially, in controlled and optimized conditions. However, fish larvae are very vulnerable during their first stages of development and undergo a severe bottleneck due to nutritional restriction, affecting their survival, and consequently, the overall fish production [2]. Newly hatched fish survive on a limited yolk supply and must encounter and successfully capture nutritionally adequate feed (e.g., zooplankton) before their energy resources become depleted [3]. Thus, adequate feeding is of pivotal importance to improve digestive capacities and ensure proper nutrition during early development to guarantee survival, growth, and health of fish larvae [4].

An emergent threat to this process is the presence of microplastics (MPs) with <300 μm in size, as they can easily resemble food items (as microalgae, zooplankton cyst/eggs) and be actively or passively captured by many planktivores, such as fish larvae. The ingestion of MPs by several fish species at larval stages have been reported in previous studies, with consequent obstruction of the gastrointestinal tract, decreased feeding activity due to a false sensation of satiation, compromised fish reproduction, and decreased larval growth [5,6]. Exposure of fish to small-sized MPs also seems to affect larval swimming capacity, induce inflammatory and metabolic responses, and affect other tissues through translocation processes [6].

The presence of MPs in aquaculture facilities is a reality, as the majority of the aquaculture systems and infrastructures (e.g., tanks and fish cages, pumps, nets, aerial traps and pots, buoyant material, among others) are based on plastic materials that are likely to degrade due to physico-chemical and biological processes [7]. In addition, hatcheries mostly rely on open systems and require water from natural environments. The presence and levels of MPs in both freshwater and marine waters is widely recognized: (1) Ma et al. [8] observed a maximal abundance of 87.5 particles/L found in Pearl River estuary of Guangzhou, China; (2) Chen et al. [9] observed the presence of MPs in mariculture on Xiangshan Bay, China, with concentrations ranging from 4.6 to 20.1 items/m$^3$ in seawater and 5570 ± 296 MPs/kg in sediments. However, these levels might be underestimated, as the majority of environmental surveys do not account for microplastics smaller than 300 μm [10]. It is predicted that 11.6–21.1 million tonnes of MPs ranging between 32–651 μm in size are suspended in the top of surface water (above 200 m) of the Atlantic Ocean [11]. Moreover, atmospheric deposition and commercial fish feeding products also constitute MPs sources to aquacultures [12,13]. Thus, an integrated understanding of how MPs may interfere with the food acquisition and processing by farmed fish larvae is necessary to apply potential mitigation measures to fulfil the nutritional requirements.

Meagre *Argyrosomus regius* has been recently identified as a promising species with high potential for the diversification of finfish aquaculture in the Mediterranean [4,14]. Using this organism as test species, this research aimed at understanding to which extent the presence of MPs affects the feeding behavior, oxidative stress status, neurotoxicity, and metabolic requirements in their larval stage, considering a real aquaculture scenario often neglected (i.e., presence/absence of food for a short-term period). For this purpose, meagre fish larvae with 15 days post-hatching (dph) were exposed to increased concentrations of polyethylene microplastics (PE-MPs) for 4 h, followed by 3 h exposure in the presence or absence of alive prey. The working hypotheses are as follows: (i) meagre larvae ingest PE-MPs; (ii) health status and feeding behavior of meagre fish larvae are affected by PE-MPs ingestion; and (iii) feed restrictions (e.g., during dark period when usually uneaten feed—live preys—are discarded) may exacerbate the potential physiological/biochemical effects of PE-MPs.

## 2. Materials and Methods

### 2.1. Organisms Culture Conditions

Fish larvae of *Argyrosomus regius* used in the assay were provided from EPPO—IPMA (Estação Piloto de Piscicultura de Olhão—Instituto Português do Mar e da Atmosfera, Portugal).

Meagre eggs were obtained from F1 broodstock (126 g viable eggs) on 21 April 2019. After the incubation period, hatched larvae (hatching rate 53.4%) were distributed into 1500 L cylindroconical tanks at a density 44 larvae/L. Since the time of mouth opening (3 dph) to 11 dph, meagre larvae were able to feed on rotifers (3–7 dph: 4 times/day, 8–10 dph: 3 times/day; 11 dph: 1 time/day). *Artemia* (EG 24 h metanauplii enriched with RedPepper, 3–4 times/day until weaning—1 time/day) and caviar (100–200 μm) were introduced after 8 and 11 dph, respectively, and *ad-libitum*. The oxygen level was 5.0 ± 0.15 mg/L, and the temperature was 20.7 ± 1.38 °C. Larvae used for testing had 15 dph, with a size-average of 4 mm.

### 2.2. Microplastic Particles Used in the Assay and Spiking

Low-density polyethylene was chosen, given its frequent detection in aquaculture [15]. Polyethylene microparticles (size: 125 μm, density 960 kg/m$^3$, shape: irregular/powder, CAS Number 9002-88-4) were purchased in Sigma-Aldrich, UK. Commercial microplastics were sieved in a vibratory sieve shaking, and particles with a size between 63–125 μm were selected and used in the bioassay, like the average size of food items provided during the rearing procedure. Three different stock solutions (15, 150, and 1500 mg/L) were prepared in filtered (0.45 μm pore size) artificial seawater (practical salinity 35; Tropic Marin® Pro Reef salt mixed with reverse osmosis water). Microplastics were allowed to age for one week. For this purpose, stock solutions were agitated continuously for one week at 50 rpm (orbital shaker) at room temperature, in the dark. Final concentrations were obtained by adding 1 mL of the respective solution to each glass containing 149 mL of seawater (final concentrations: 0.1, 1.0, 10 mg/L). In control conditions (0 mg/L MPs), a volume of 1 mL of artificial seawater (also left to shake in the same conditions that MPs stock solution) was added to each glass vial of such treatment. Such high PE-MPs concentrations were chosen to stimulate larvae to initiate feeding during the relatively short experimental period and infer potential thresholds on feeding behavior and physiological/biochemical endpoints (as further explained in the next section).

### 2.3. Experimental Setup

*Argyrosomus regius* larvae (15 dph) were exposed for 4 h to the three PE-MP concentrations (0.1, 1, and 10 mg/L) and a control treatment (0 mg PE-MPs/L). This period was followed by an additional 3 h exposure, where half of the treatments were fed with live prey (5 *Artemia* metanauplii/mL), and another half remained without food supply. Thus, a total of eight treatments were prepared according to Figure 1.

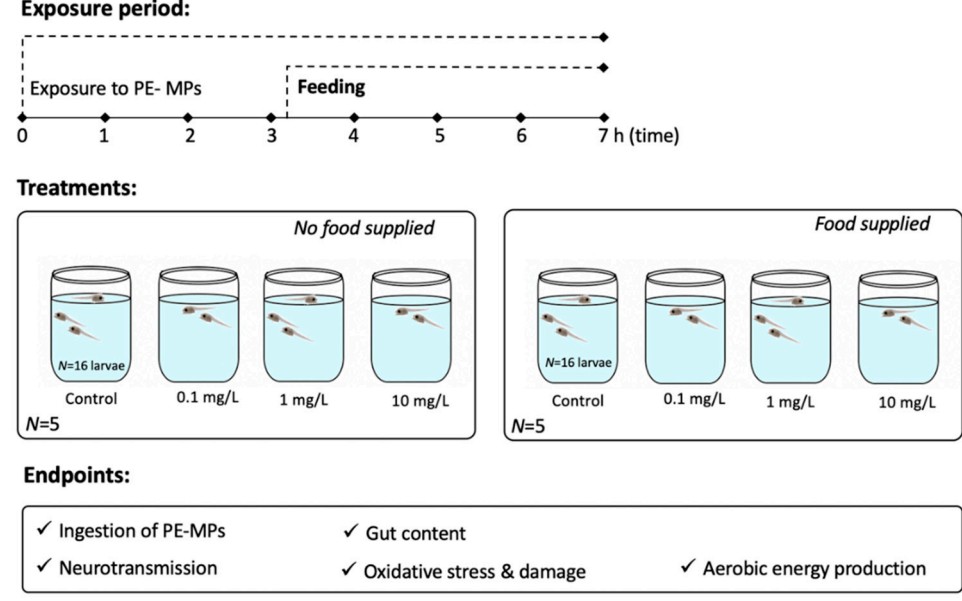

**Figure 1.** Schematic representation of the experimental setup and target endpoints.

Briefly, each treatment consisted of five glass vials, each containing 150 mL of test medium and 16 meagre larvae (although corresponding to a density of 107 larvae/L, it does not compromise meagre larvae survival as in Roo et al. [16]). The experiment was conducted at controlled temperature of 19 ± 1 °C with no aeration. Water parameters were monitored at the beginning and the end of the experiment. After exposure, two pools of three fish larvae were anaesthetized with 1% 2-phenoxyethanol, gently rinsed with Milli-Q ultra-pure water. Then one pool was preserved in 70% ethanol to assess the ingestion of PE-MPs, and the other pool was fixed with 4% formaldehyde to assess gut content. The

remaining fish larvae were sensed with Milli-Q ultra-pure water, gently dried with filter paper, placed into 2 mL Eppendorf tubes, and deep-frozen in liquid nitrogen to assess biochemical responses.

### 2.4. Extraction and Quantification of Microplastics

Before extraction and quantification of MPs, all larvae were (once more) gently rinsed and observed under the stereomicroscope to verify the absence of PE-MPs adhered to their skin. The PE-MP extraction followed the optimized protocol developed by Silva et al. [17], whereas the quantification method followed Prata et al. [18]. Briefly, to assess the presence of PE-MPs inside organisms, larvae were placed in glass flasks, covered with aluminum foil, and dried at 50 °C for 24 h. Afterwards, 3 mL of nitric acid ($HNO_3$, 65%) was added to each replicate and incubated at 60 °C for three hours. After this period, samples were allowed to cool down to room temperature (RT), and 2.6 mL of hydrogen peroxide ($H2O_2$, 35%) were added. Samples were incubated at RT overnight. Then, samples were diluted with 50 mL of Milli-Q ultra-pure water and immediately vacuumed filtered onto black polycarbonate filters (PCTE, 0.2 μm pore size, 42 mm ∅, ref. 7063–4702, Cytiva WhatmanTM, Fisher Scientific, Portugal) to retain PE-MPs. Microplastics were stained with 1 mL of Nile red (Sigma Aldrich, St. Louis, MO, USA; stock solution: 0.01 mg/mL ethanol absolute) for 5 min protected from the light [18]. After that, filters were washed with ultrapure water to remove the excess dye and stored in glass Petri-dishes. After drying, polycarbonate filters of each sample were photographed (Canon 550D, EF-S 18–55 mm, Oita, Japan) under a blue light (450 nm, SPEX Forensics, USA) in a dark room using an orange filter (Standard ProMaster®), and the number of microplastics was counted.

Several measures were taken to ensure quality criteria and quality control on microplastics analysis. Glassware (thoroughly acid-washed and rinsed with Milli-Q ultra-pure water) was preferential for testing and analysis; samples were prepared for digestion under a clean laminar chamber and covered with aluminum foil to avoid airborne contamination. During extraction and quantification procedures, blanks (1 every 5 samples) were prepared to address possible cross-contamination between samples.

### 2.5. Gut Content Analysis

To evaluate the gut content, larvae of each treatment (previously kept in 4% formaldehyde) were transferred to ethanol (95%) for 2 h, and after that, moved to potassium hydroxide (1%) for 1 day. Then, the number of preys in fish larvae gut was quantified using a binocular microscope, after opening the fish gut with a micro scissor and extracting the content with a dissecting needle [19]. The feeding activity was expressed as the number of ingested preys per larvae.

### 2.6. Oxidative Stress, Aerobic Energy Production, and Neurotoxicity

The biochemical responses were determined following the previously optimized protocols to microplates [20–23]. Briefly, each sample was homogenized in 1000 mL of ultra-pure water and on ice by sonication (30 s at a pulse mode of 60%, 250 Sonifier, Branson Ultrasonics). Further, three aliquots of 300, 150, and 50 μL for ETS—electron transport system, LPO—lipid peroxidation, and protein quantification, respectively, were set aside. To the remaining volume (500 μL), 0.2 M K-phosphate buffer, pH = 7.4, (1:1) was added, and samples were centrifuged at $10,000 \times g$ at 4 °C for 20 min. Posteriorly the resultant post-mitochondrial supernatant (PMS) was divided into aliquots for CAT—catalase, GST—glutathione-*S*-transferase, and ChE—cholinesterase activities determination, and tGSH—total glutathione and protein levels quantification.

To evaluate oxidative damage, lipid peroxidation was measured using Thiobarbituric acid reactive substances (TBARS) assay [24,25]. The absorbance was read at 535 nm. Using a molar extinction coefficient ($\epsilon$) = $1.56 \times 105$ $M^{-1}$ $cm^{-1}$, the results were expressed as nmol TBARS/mg of protein. Catalase activity was measured in 20 μL of PMS and determined, according to Clairborne [26]. The consumption of the hydrogen peroxide

($H_2O_2$) was read at 240 nm for 2 min, and the results were presented as μmol/min/mg of protein, using a $\epsilon = 40$ $M^{-1}$ $cm^{-1}$. The activity of detoxification enzyme GS T was determined (50 μL of PMS), according to Habig et al. [27], by measuring the formation of glutathione dinitrobenzene (340 nm) for 3 min. Results were expressed as nmol/min/mg of protein using a $\epsilon = 9.6 \times 103$ $M^{-1}$ $cm^{-1}$. Effects on neurotransmission were evaluated by measuring the ChE activity, following the method described by Ellman et al. [28], Guilhermino et al. [29], and Domingues and Gravato [30]. Using acetylthiocholine as substrate and 50 μL of PMS, the absorbance was read 412 nm for 5 min, and the results were expressed as nmol/min/mg of protein using a $\epsilon = 13.6 \times 103$ $M^{-1}$ $cm^{-1}$. Total glutathione levels were calculated in 50 μL of PMS, according to Baker et al. [31]. The absorbance was read at 412 nm for 3 min following the recycling reaction of reduced glutathione in excess of glutathione reductase. The results were expressed as μM per mg of protein, using a standard curve with L-GSH as a standard as previously performed by Campos et al. and Rodrigues et al. [32,33]. Effects on aerobic energy production were evaluated determining the ETS activity according to De Coen and Janssen [34] with some adaptations [23].

Protein levels of each sample were determined following Bradford's method, and the absorbance was read at 592 nm. Bovine serum albumin was used as the standard for quantification [35]. The protein of PMS was used to calculate CAT, GST, ChE activities, and tGSH levels, while total protein was used to calculate ETS activity and LPO.

### 2.7. Statistical Analysis

Significant effects on the feeding activity of *A. regius* larvae after exposure were evaluated through one-way analyses of variance (ANOVA). Polyethylene microplastics ingestion and biochemical responses were evaluated by two-way ANOVA being PE-MPs concentration one factor and presence/absence of food another factor. The control condition (0 mg/L of PE-MP) was excluded in the analysis of microplastics ingestion since no ingestion is expected/occurs in the absence of MPs. The normality of all variables was assessed on residues using the Shapiro–Wilk test, while Levene's test verified the homoscedasticity of data ($p > 0.05$). Dunnett's test was used to determine significant differences from respective controls ($p < 0.05$). All the statistical analysis and graphical representation of the results were performed using GraphPad Prism 8 (GraphPad Software, San Diego, CA, USA).

## 3. Results and Discussion

The temperature, pH, and salinity remained stable during the exposure period (monitored at the beginning and the end of the bioassay), temperature: $19.1 \pm 0.14$ °C; pH: $7.70 \pm 0.01$; salinity: 35. The dissolved oxygen decreased over time as no aeration was provided, but it remained above the limit of 5 mg/L ($5.4 \pm 0.1$). The average fish larvae survival in control, 0.1, 1, 10 mg PE/L treatments was, respectively, 96.25%, 98.44%, 100.00%, and 97.50%; in treatments with live preys added; and 87.50%, 93.75%, 96.89%, and 95.00% in treatments with no feed added.

### 3.1. Ingestion of Polyethylene Microplastics (PE-MPs) and Prey Items by Meagre Fish Larvae

Larvae exposed to PE-MPs presented MPs particles in their gut. The number of PE-MPs in the larval gut was not dependent on the tested concentration ($F_{(2, 23)} = 2.924$, $p = 0.074$, Table 1), but it was significantly lower in the presence of live prey ($F_{(1, 23)} = 5.351$, $p = 0.030$). Although, no interaction was observed between both factors ($F_{(2, 23)} = 0.5364$, $p = 0.592$).

The observed low numbers of PE-MPs in the larval fish gut (independent of PE-MPs concentration) suggest unintentional intake followed by intentional elimination (or intentional spitting). Since meagre larvae possess a gut capacity to accumulate 8–15 *Artemia* prey of 500–800 μm in size (as observed in Figure 2—control conditions), they had the potential for accumulating at least a similar number of PE-MPs than prey, particularly under starvation (i.e., in treatments where no prey was provided for 7 h). In addition,

PE-MPs tested concentration were considerably higher (in the number of particles) than the total number of the prey supplied to fish larvae; thus, an unintentional capture and retention for "digestion" (i.e., without spitting) would result in a higher number of PE-MPs in larval gut. Yet, this was not the case with meagre larvae. In fact, some replicates (which consisted of a pool of three larvae) did not present any PE-MPs, or presented an average number < 1 particle per larvae.

**Table 1.** Number of polyethylene microplastics (PE-MPs) ingested by *Argyrosomus regius* larvae, in the presence and absence of live prey (*Artemia* metanauplii). Data are presented as mean ± standard error of the mean.

| [PE-MPs] mg/L | without Food | with Food |
|---|---|---|
| 0 | - | - |
| 0.1 | 1 ± 0.2 | 0.2 ± 0.3 |
| 1 | 1.2 ± 0.1 | 0.8 ± 0.2 |
| 10 | 1.3 ± 0.4 | 1.1 ± 0.2 |

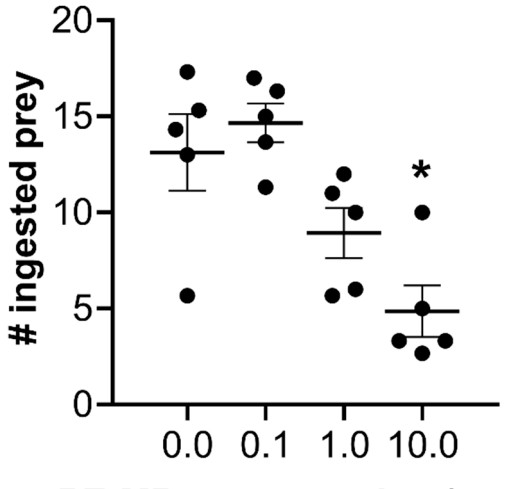

**Figure 2.** Number of *Artemia* nauplii (prey) ingested by *Argyrosomus regius* larvae exposed to polyethylene microplastics (PE-MPs) after 3 h feeding period. Data are presented as mean ± standard error of the mean. (*) denote significant statistical differences compared to control condition, 0 mg PE-MPs/L (Dunnet's post-hoc test, $p < 0.05$).

Microplastics' ingestion was expected, as these particles can resemble planktonic organisms or particulate food items in size (65–125 μm) and shape (irregular to bead shape); thus, becoming available to be captured by many planktivores such as fish larvae. Nevertheless, most fish larvae also use visual and chemical (odor, palatability) stimuli to detect suitable food or egest indigestible items [36]. Considering obtained results, meagre fish larvae seem to rely on chemical stimuli to recognize PE-MP as inedible materials and then spit them, after their active or passive ingestion. Such behavior was observed in zebrafish (*Danio rerio*), which seemed to ingest PE particles quickly, followed by egestion [37]. Low number of MPs particles was equally found in the European white fish larval gut (*Coregonus lavaretus*, dph not mentioned), which presented 0–6 particles of 90 μm polystyrene beads per larvae (concentration tested up to 30 beads/mL) after 2–6 h exposure [38], and also in seabass fish larvae (*Dicentrarchus labrax*, 7–43 dph), which presented 1.4 to 3.3 particles of 10–45 μm PE microbeads per larval gut (concentration tested up to 12 mg/L) [39]. Moreover, recent studies observed that the abundance of MPs inside fish larvae is related to feeding type, the ingestion of MPs being lower in carnivorous/planktivorous comparatively to omnivores and filter feeders [40,41].

The exposure to PE-MPs affected the number of prey ingested/digested by fish larvae, with a lower number of prey found in larvae exposed to the highest tested concentration compared to the control condition ($F_{(3, 16)}$ = 9.161, $p$ = 0.001) (Figure 2).

Fed larvae did not reveal high(er) levels of PE-MPs in their gut that could have occupied gut volumetric space (Table 1). The low number of prey found in fish could be due to four potential reasons: (i) in the presence of MPs, fish larvae potentially spit more particles from their mouth to promote water flux through gills; thus, food items would also be excluded from their mouth; (ii) alteration of gut microbiota and/or irritability/damage of the gut tissues after ingestion/egestion of MPs (although the number of ingested PE-MP remains low after 7 h, it is unclear how many particles they ingested/spit during that period); (iii) a decrease in larvae locomotion/swimming activity due to neurotoxicity or oxidative stress induced by PE-MP exposure/ingestion (as discussed below); and (vi) an increase in prey digestion processing induced by the ingestion/egestion of PE-MPs (less likely, as prey remained mostly undigested at the sampling time).

Although the numbers of ingested PE-MPs were generally low, with meagre fish revealing the potential capacity to egest (spit) the majority of them, the presence and ingestion of MPs may still represent a significant threat in aquaculture facilities where the range of sizes, shapes, and chemical compositions is wider. The effects are also likely to vary spatially, as MPs density relies on environmental variables (e.g., flow, volume capacity). In addition, most aquaculture facilities are typically located in littoral zones, where high densities of MPs have been reported as a result of photooxidation and mechanical degradation of macro- and microplastics, or a result of WWTP discharges [42,43].

### 3.2. Effects of PE-MPs Exposure and Food Supply on the Biochemical Responses of Meagre Larvae

The presence and ingestion of PE-MPs induced several biochemical responses on meagre fish larvae, as depicted in Figures 3–5 and Table 2 (for statistical support). In our study, although the presence/absence of food altered the biochemical responses of organisms, in general, it seems not to change the effects of MPs since no interaction on the two-way ANOVA was observed to all endpoints evaluated but lipid peroxidation (LPO, Table 2).

Results show a generalized increment (up to 81 and 101%, in the absence and presence of food, respectively) on glutathione levels (tGSH) of meagre larvae exposed at concentrations $\geq$ 1 mg PE-MPs/L (Figure 3a, Table 2). Moreover, significant inhibition of catalase activity (up to 31% and 48% in the absence and presence of food, respectively) was observed in all tested concentrations in the presence of food and at 0.1 and 1 mg PE-MP/L in the absence of food (Figure 3b, Table 2).

Such results are underlying the activation of the antioxidant system to deal with the generation of reactive oxygen species (ROS). ROS generation could be a result of, for instance, an immune response triggered after the ingestion of PE-MPs as a consequence of chemical stimuli or a potential mechanical damage/abrasion or proteolytic damage to the epithelial cells of the gut lumen of fish larvae. Activation of an immune response was also previously inferred by upregulated genes in zebrafish larvae (*D. rerio*) after exposure to polyethylene and polystyrene microplastics [44]. Although CAT is considered an enzyme of the first line of defense that directly eliminates ROS, its activity is regulated by other players in the antioxidant system, such as glutathione peroxidase, superoxide dismutase, and even total glutathione levels, and relies on the energy available for their activation [45]. In our case, CAT inhibition was (indeed) accompanied by an increment of tGSH, which seemed to counteract the potential oxidative damage, as observed by the absence of an increase in lipid peroxidation (LPO, Figure 3c, Table 2). The triggering of glutathione system to prevent oxidative damage, namely de novo synthesis and recycling of reduced GSH [46], was also observed in other marine invertebrates such as rotifers (*Brachionus koreanus*) and crabs (*Eriocheir sinensis*) exposed to MPs (as reviewed in [47]). Glutathione has been highlighted as one of the key components of the antioxidant system in preventing oxidation of lipids and proteins by various environmental xenobiotics, including MPs [47–49].

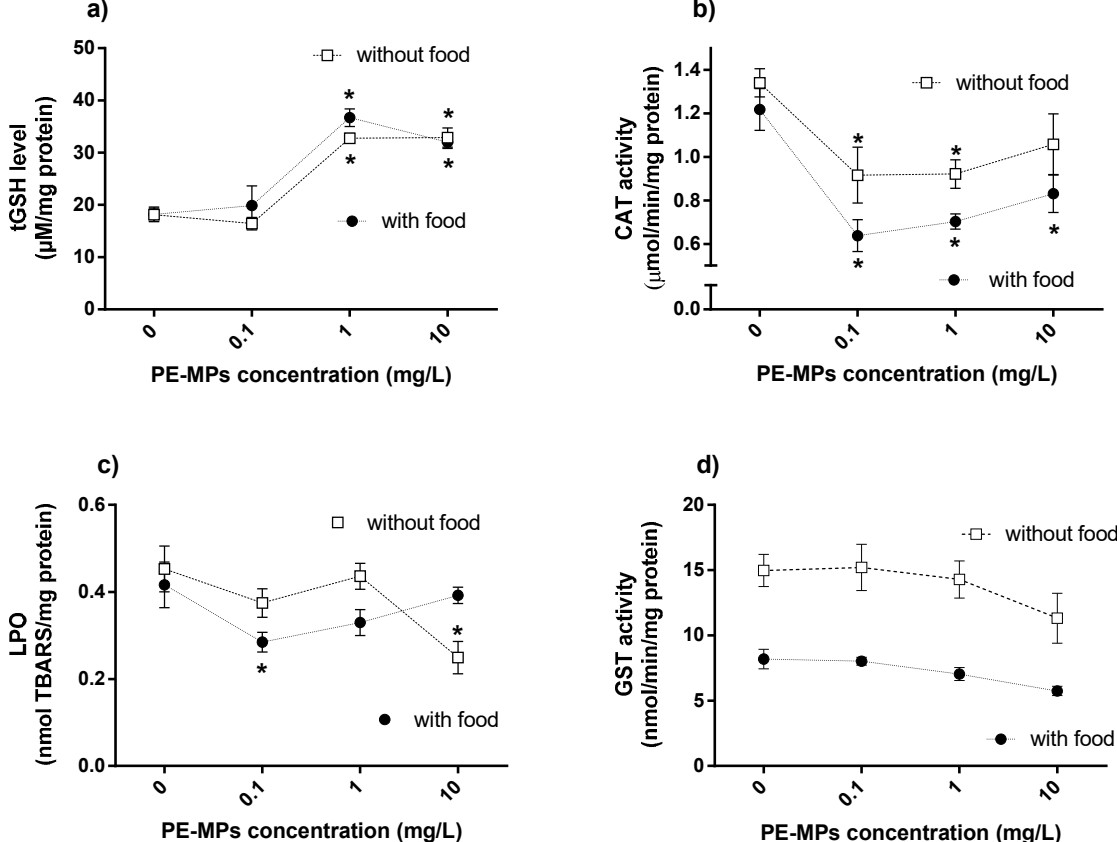

**Figure 3.** Biochemical responses of *Argyrosomus regius* larvae after 7 h exposure to polyethylene microplastics (PE-MPs), in the presence or absence of food, on (**a**) catalase activity (CAT), (**b**) total glutathione levels (tGSH), (**c**) glutathione-*S*-transferase activity (GST) and (**d**) Lipid peroxidation (LPO) in the presence (black circle) and absence (white square) of food. Data is presented as mean ± standard error of the mean. (*) denote significant statistical differences compared to respective control, 0 mg PE-MPs/L (Dunnett's post-hoc test, $p < 0.05$).

Exposed meagre larvae also revealed an increase (up to 35%) in the aerobic energy production, particularly at 1 and 10 mg/L PE-MPs for starved larvae, and at 0.1 and 1 mg/L of PE-MP on fed larvae (Figure 4). Such aerobic energy production could have been allocated to the generalized synthesis and recycling of GSH to combat ROS and oxidative damage. In fact, exposed larvae revealed a decrease in LPO levels, particularly under starvation, in the highest tested PE-MPs concentration (10 mg/L) (Figure 3c, Table 2). Such a decrease in LPO could be a result of the increase in tGSH levels, and a concomitant decrease in lipid reserves (although not measured, but expected to be mobilized under starvation/ingestion of PE-MPs by meagre fish larvae that also possesses a considerable fast growth) [50]. Fed fish larvae were able to obtain and replenish some of the lipid reserves, which could explain the relatively stable LPO in most of PE-MPs concentration (Figure 3c, Table 2).

Despite exposure to, and/or ingestion of, PE-MPs triggered responses from tGSH, it seems not to significantly alter GST activity (Figure 3d, Table 2), suggesting no alteration of this enzyme biotransformation action, which was already observed in *Pomatoschistus microps* exposed up to 185 µg/L microspheres (1–5 µm) [51]. Nevertheless, meagre fish under starvation presented higher GST activity than larvae fed with *Artemia* microcrustaceans (Figure 3d, Table 2). These results are not surprising, since food deprivation is also a known stressor to fish with the potential to induce ROS generation and oxidative stress, which involves GST, as observed in the juveniles of Siberian sturgeon (*Acipenses baerii*) and European sea bass (*Dicentrarchus labrax*) after starvation periods [50,52].

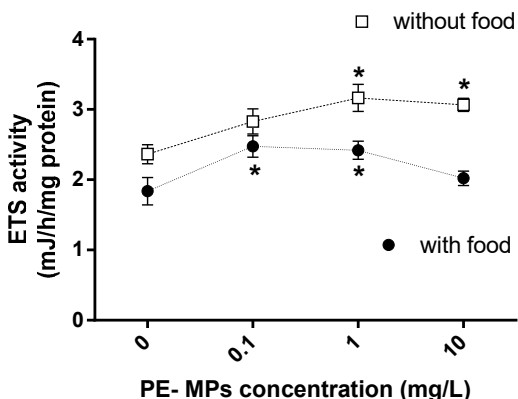

**Figure 4.** Responses of *Argyrosomus regius* larvae after 7 h exposure to polyethylene microplastics (PE-MPs) on electron transport system activity (ETS) in the presence (black circle) and absence (white square) of food. (*) denote significant statistical differences compared to respective control, 0 mg PE-MPs/L (Dunnett's post-hoc test, $p < 0.05$).

**Table 2.** Two-way ANOVA results testing for concentrations of polyethylene microplastics [PE-MPs], presence or absence of food (p/a food), and interaction between both on lipid peroxidation (LPO), catalase activity (CAT), glutathione-*S*-transferase activity (GST) total glutathione contents (tGSH), aerobic energy production (ETS—electron transport system) and cholinesterase activity (ChE).

|  |  | **Sums of Squares** | **F-Value, *df*** | ***p*-Value** |
|---|---|---|---|---|
| **LPO** | [PE-MPs] | 0.084 | $F_{(3,32)} = 4.207$ | **0.013** |
|  | p/a food | 0.005 | $F_{(1,32)} = 0.7594$ | 0.390 |
|  | Interaction | 0.099 | $F_{(3,32)} = 4.919$ | **0.006** |
| **CAT** | [PE-MPs] | 1.57 | $F_{(3,32)} = 12.40$ | **<0.001** |
|  | p/a food | 0.45 | $F_{(1,32)} = 10.61$ | **0.003** |
|  | Interaction | 0.03 | $F_{(3,32)} = 0.2496$ | 0.861 |
| **GST** | [PE-MPs] | 62.81 | $F_{(3,32)} = 2.952$ | **0.047** |
|  | p/a food | 448.30 | $F_{(1,32)} = 63.21$ | **<0.0001** |
|  | Interaction | 4.524 | $F_{(3,32)} = 0.2126$ | 0.887 |
| **tGSH** | [PE-MPs] | 2407 | $F_{(3,32)} = 46.88$ | **<0.0001** |
|  | p/a food | 27.55 | $F_{(1,32)} = 1.610$ | 0.214 |
|  | Interaction | 43.74 | $F_{(3,32)} = 0.8519$ | 0.476 |
| **ETS** | [PE-MPs] | 2.68 | $F_{(3,32)} = 7.756$ | **0.0005** |
|  | p/a food | 4.47 | $F_{(1,32)} = 38.76$ | **<0.0001** |
|  | Interaction | 0.66 | $F_{(3,32)} = 1.919$ | 0.146 |
| **ChE** | [PE-MPs] | 9037 | $F_{(3,32)} = 25.16$ | **<0.0001** |
|  | p/a food | 862.5 | $F_{(1,32)} = 7.204$ | **0.0114** |
|  | Interaction | 998.4 | $F_{(3,32)} = 2.779$ | 0.0570 |

Regarding cholinergic neurotransmission, the present results showed a decrease in the cholinesterase activity of *A. regius* larvae after only 7 h exposure to all tested PE-MPs (even environmentally relevant [10]) in the presence and absence of food (Figure 5, Table 2). Cholinesterase has a vital role in the transmission of impulses between neurons by hydrolysing the acetylcholine into choline in synaptic clefts. The consequence of cholinesterase inhibition is the accumulation of neurotransmitters in the synaptic cleft and disturbance of nerve impulse transmission. The observed inhibition rate of ChE (up to 25%) has been considered high enough to induce adverse effects in neurofunction in other exposed organisms, including fish [53], among which motor, sensory, or cognitive activities are highly affected. The mechanisms behind ChE inhibition remain unclear, but several hypotheses have been raised. For instance, exposure to MPs has the potential to increase reactive oxygen species proliferation and oxidative damage, which can inclusively affect

the cells of the nervous system and, thus, neurotransmission [53]. The search for food, and the ingestion of indigestible particles such as MPs, seem to impose developmental costs to meagre fish larvae, by affecting energy assimilation and increasing energetic costs (for detoxification), which can ultimately lead to a decrease in organism activity and eventually neurotransmission [54,55]. ChE inhibition has also been theoretically related to plasticizers or unknown hazardous chemicals released by MPs and to non-cholinergic functions related to neurite growth, synaptogenesis, cell migration, proliferation, and cell apoptosis [53]. In our case, meagre fish larvae exposed to pristine PE-MPs revealed an increment in glutathione levels and increased energy consumption to reduce ROS and avoid oxidative damage. Thus, the inhibition of ChE is potentially related to such readjustments on antioxidant systems and aerobic metabolism that could have affected fish larvae behavior (i.e., decrease in swimming activity—although not measured) and reduce cholinergic transmission. ChE inhibition can also be related to non-cholinergic functions or other parallel toxic mechanisms, although these mechanisms remain to be determined in the presence of MPs. Regardless of the mechanisms behind neurotoxicity, ChE inhibition in fish would be alarming, taking into consideration the ubiquity of MPs in the aquatic environment and the pivotal role of this enzyme in neurological function, which is crucial to control several physiological (e.g., growth, reproduction) and behavioral (e.g., swimming) processes that directly or indirectly may influence individual and population fitness.

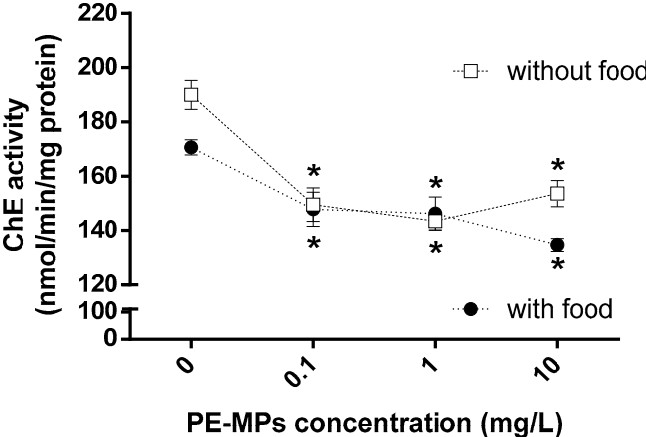

**Figure 5.** Responses of *Argyrosomus regius* larvae after 7 h exposure to polyethylene microplastics (PE-MPs) on Cholinesterase activity (ChE) in the presence (black circle) and absence (white square) of food. (*) denote significant statistical differences compared to respective control, 0 mg PE-MP/L (Dunnett's post-hoc test, $p < 0.05$).

## 4. Concluding Remarks

Our results provide the first evidence of the potential adverse effects of MPs on finfish hatcheries under realistic scenarios (i.e., presence and absence of food). Meagre fish larvae seems to be able to ingest and spit PE-MPs, however with metabolic requirements and energetic costs associated. The increment of non-enzymatic antioxidant defenses and aerobic energy production, in the present study, revealed that fish larvae were able to cope with the increase of ROS triggered by the ingestion and presence of MPs and prevent oxidative damage. On the other hand, farmed organisms are frequently experiencing feeding restrictions during several periods, namely, night period, to monitor water quality and prevent diseases. Consequently, and considering the presence of MPs in aquaculture systems, their toxic effects could be enhanced in the periods of absence of food.

In conclusion, our results showed that, although organisms had the ability to cope with stress (at short-term exposure to PE-MPs), the effects observed can compromise the success/efficiency of aquaculture production, because (i) MPs triggered biochemical responses on vulnerable early life stages—which can lead to an inefficient finfish growth and development at short-term and (ii) cumulative long-term effects that cannot be ruled out could

threaten the organisms' health and life traits. Moreover, the effects of microplastics can differ due to the variety of polymeric composition, size, shape, and plasticizers/adsorbed contaminants. Thus, the presence of MPs and their vector nature should be deeply studied and understood in aquaculture scenarios, and mitigation strategies should be adopted to prevent such contaminations.

**Author Contributions:** Conceptualization and visualization: A.L.P.S., D.C., A.C.M.R., R.J.M.R., and R.M.; methodology: D.C. and A.C.M.R.; validation: A.L.P.S., D.C., A.C.M.R., R.J.M.R., R.M., and C.G.; formal analysis: A.L.P.S., D.C., A.C.M.R., R.J.M.R., R.M., and C.G.; resources: A.C.-M., S.C., F.S., P.P.-F., A.L.P.S., and R.J.M.R.; data curation: A.L.P.S., D.C., A.C.M.R., and C.G.; writing—original draft preparation: A.L.P.S., D.C., A.C.M.R., and C.G.; writing—review and editing: A.L.P.S., D.C., A.C.M.R., R.J.M.R., R.M., C.G., A.C.-M., S.C., F.S., P.P.-F., and A.M.V.M.S.; supervision: A.L.P.S., R.J.M.R., F.S., P.P.-F., and A.M.V.M.S.; project administration: A.L.P.S., R.J.M.R., A.M.V.M.S., F.S., and P.P.-F.; funding acquisition: A.L.P.S., R.J.M.R., A.M.V.M.S., and P.P.-F. All authors have read and agreed to the published version of the manuscript.

**Funding:** This research was funded through CESAM (UIDP/50017/2020+UIDB/50017/2020), with the financial support from FCT/MCTES through national funds; to the research project ComPET (PTDC/CTA-AMB/30361/2017) funded by FEDER, through COMPETE 2020—Programa Operacional Competitividade e Internacionalização (POCI), and by national funds (OE); to the research project Diversiaqua II (Mar2020-P02M01-0656P) supported by MAR2020 program; and to the Integrated Program of SR&T D' Smart Valorisation of Endogenous Marine Biological Resources Under a Changing Climate' (reference Centro-01-0145-FEDER-000018), co-funded by Centro (2020) program, Portugal 2020, European Union, through the European Regional Development Fund. A.L.P.S. and R.M. were supported by CEECIND/01366/2018 and CEECIND/01329/2017, respectively. D.C. has a research contract within the project ComPET (nr. 5662).

**Institutional Review Board Statement:** The current research was led by accredited Researchers by the Portuguese National Authority for Animal Health (DGAV—Direção Geral de Alimentação e Veterinária) for the practice of animal experimentation (FELASA Category B), following the Portuguese Law nr. 113/2013 and the European Directive 2010/63/EU of the European Parliament and the European Union Council on the protection of animals used for scientific purposes.

**Informed Consent Statement:** Not applicable.

**Data Availability Statement:** The data presented in this study is available in the current manuscript, raw data is available on request from the corresponding author.

**Acknowledgments:** We would like to thank to Sara Silva, Lígia Santana, and Joana Figueiredo for the valuable help when handling the fish larvae for testing.

**Conflicts of Interest:** The authors declare no conflict of interest.

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
