# Peer review of "Are Microplastics Impairing Marine Fish Larviculture?—Preliminary Results with Argyrosomus regius"

_water, doi:10.3390/w13010104_

Round 1
Reviewer 1 Report
Water-1040398
Title: Are microplastics impairing marine fish larviculture? - preliminary results with Argyrosomus regius
This is very interesting and timely research and organized well. I suggest some minor modifications. Please see below comments.
- Methods
- (Line 95) 2.2 – Authors used LDPE in this test, and I guess artificial seawater is much heavier than this MPs and MPs might be floating on the water surface. How did authors control them? Or just let them be floating on the water surface?
- (Line 100) 2.2 – Please use SI unit of the salinity.
- (Lines 101-102) 2.2 – Didn’t authors use any surfactants or other chemicals to make stock solutions except shaking? I guess that makes some heterogeneity of MP in the stock solution.
- (Line 106) – 2.2 “food-naïve” in English?
- (Line 142) – 2.4 “Red Nile” isn’t it “Nile red”?
- (Line 143 – 144) 2.4 – Authors washed filters after staining, but I wonder isn’t there some lost of MPs in this process.
- Results
- (Line 205) 3 – Please use SI unit of the salinity.
- (Line 207) 3 – Authors conducted this experiment for 7 hours to confirm the relationships between MP ingestion and food existence in the environment. However, the average survival rate of fish larvae, 87%, is too low. Is there any reason for this? And I suggest that authors suggesting the survival rate in each treatment in the manuscript.
Author Response
This is very interesting and timely research and organized well. I suggest some minor modifications. Please see below comments.
Methods(Line 95) 2.2 – Authors used LDPE in this test, and I guess artificial seawater is much heavier than this MPs and MPs might be floating on the water surface. How did authors control them? Or just let them be floating on the water surface?
Authors: We understand the reviewer concern. PE-MPs have a low density (917-960 kg/m3) and, without any ageing process, the particles tend to float. To overcome this, PE-MPs were allowed to “age” in test-medium for 1 week before experiments. With this procedure, PE-MP particles remained mainly in the water column (although some larger size particles could remain on the surface). Fish swimming activity also contributed to the permanence of PE-MPs in the water column (as observed by naked eye).
We rephrased for clarity:
LL 101-103: “Microplastics were allowed to age for one week. For this purpose, stock solutions were agitated continuously for one week at 50 rpm (orbital shaker) at room temperature, in the dark.”
(Line 100) 2.2 – Please use SI unit of the salinity.
Authors: We followed the recommendation of UNESCO PSS-78: practical salinity is a dimensionless parameter and should be represented only by a number (e.g. S=35).
(Please see https://www.nature.com/scitable/knowledge/library/key-physical-variables-in-the-ocean-temperature-102805293/ )
It can now be read:
LL 99 -101: “Three different stock solutions (15, 150, 1500 mg/L) were prepared in filtered (0.45 µm pore size) artificial seawater (practical salinity 35; Tropic Marin® Pro Reef salt mixed with reverse osmosis water).”
(Lines 101-102) 2.2 – Didn’t authors use any surfactants or other chemicals to make stock solutions except shaking? I guess that makes some heterogeneity of MP in the stock solution.
Authors: Microplastics of this size do not present colloidal behaviour; thus, we did not apply any surfactant. Surfactant itself could also have toxic effects to fish larvae. PE-MPs were only prepared in test medium (seawater). No heteroaggregation or heterodispersion was observed.
(Line 106) – 2.2 “food-naïve” in English?
Authors: We rephrased for clarity:
LL 106-109: “Such high PE-MPs concentrations were chosen to stimulate larvae to initiate feeding during the relatively short experimental period and infer potential thresholds on feeding behaviour and physiological/biochemical endpoints (as further explained in the next section).”
(Line 142) – 2.4 “Red Nile” isn’t it “Nile red”?
Authors: Altered as suggested. Thanks.
(Line 143 – 144) 2.4 – Authors washed filters after staining, but I wonder isn’t there some lost of MPs in this process.
Authors: The filters were washed in the glass vacuum system, after staining to remove the excess of Nile red, since this could difficult/ interfere with the counting of MPs. There is a chance of loosing particles during the process (although not by the washing process as the filter pore was considerably lower than the particle size). Nevertheless, and based in our experience when optimizing this digestion procedure, we have a high percentage of digestion efficiency and high recovery rate for MPs (> 90%, unpublished data). Particles loss can be related to the adherence of MPs (after extraction) to the glass vial walls. But several washing procedures with filtered UP water help recovering most of them.
Results, (Line 205) 3 – Please use SI unit of the salinity.
Authors: As previously replied, we use practical salinity without units.
(Line 207) 3 – Authors conducted this experiment for 7 hours to confirm the relationships between MP ingestion and food existence in the environment. However, the average survival rate of fish larvae, 87%, is too low. Is there any reason for this? And I suggest that authors suggesting the survival rate in each treatment in the manuscript.
Authors: We now provide the mean survival percentage for all treatments. Previously, data was presented as the lowest survival value observed per replicate (≥87%). Meagre larvae are sensitive to handling procedures, at this development stage, and this might be contributed for the random mortality observed.
We rephrase for clarity:
LL 209-211: “The average fish larvae survival in control, 0.1, 1, 10 mg PE/L treatments was 96.25, 98.44, 100.00 and 97.50 %; in treatments with live preys added; and 87.50, 93.75, 96.89 and 95.00 % in treatments with no feed added.”
Reviewer 2 Report
The manuscript reports experimental evidence of the possible effects of MPs in aquaculture facilities. The study is interesting especially in light of the great importance of aquaculture in the future blue economy. The paper is well organized and clearly presented.
I have only few questions for the Authors and few minor comments.
The Authors have tested only low-density polyethylene, it would have been interesting to test different polymers. Most importantly, they have tested the pure polymer, whereas the effects of chemical additives normally present in the plastic production could play a role. Do the Authors have a comment on this?
The Authors should mention similar work that are published on the toxicity of MPs on several species.
Minor comments:
Figure and tables should be improved, especially figure 1.
Based on my previous comments the manuscript is acceptable for publication.
Author Response
The manuscript reports experimental evidence of the possible effects of MPs in aquaculture facilities. The study is interesting especially in light of the great importance of aquaculture in the future blue economy. The paper is well organized and clearly presented.
I have only few questions for the Authors and few minor comments.
The Authors have tested only low-density polyethylene, it would have been interesting to test different polymers. Most importantly, they have tested the pure polymer, whereas the effects of chemical additives normally present in the plastic production could play a role. Do the Authors have a comment on this?
Authors: We appreciate the reviewer’s comment. Our aim was to address the effects of microplastics per se on fish larvae in their early stages and when they start to feeding and consequently prone to ingest such particles. We used polyethylene (PE) particles as a first approach as it is the most common polymer found in marine environments and aquaculture facilities, as stated in the Methods subsection 2.2.
The tested PE particles were pristine and free of additives (as we wanted), as confirmed by a FTIR-ATR spectrum; thus, the effects observed were mostly related to the polymer size/shape/composition rather that their plasticizers leachates.
In natural environments (and due to their small size, high surface/volume ratio, and hydrophobicity) microplastics are likely to absorb chemicals, but this was not the aim at this first stage (address the behaviour of MPs as vectors).
However, we agree with the reviewer and further studies could encompass other polymers (size/shape/composition) and their nature as vectors.
We emphasized this concern in the conclusion section, and the text can now be read:
LL 409-412: “Moreover, the effects of microplastics can differ due to the variety of polymeric composition, size, shape, and plasticizers/adsorbed contaminants. Thus, the presence of MPs and their vector nature should be deeply studied and understood in aquaculture scenarios, and mitigation strategies should be adopted to prevent such contaminations.”
The Authors should mention similar work that are published on the toxicity of MPs on several species.
Authors: We understand the reviewer concern. However, and for a more integrative discussion, we decided to narrow the literature to papers published on fish larva with similar developmental stage (i.e., when they start feeding on live preys) and short-term exposures.
Minor comments:
Figure and tables should be improved, especially figure 1.
Authors: We improved the quality of figures and table format. Thanks.
Based on my previous comments the manuscript is acceptable for publication.
Reviewer 3 Report
In the present study the authors examined the short-term effects of small-sized microplastics in aquaculture facilities which may threaten finfish hatchery and may compromise their nutritional requirements during early ontogeny, affecting consequently larval health and performance. Through fasting and feeding fish trials, they evaluated important endpoints (larval feeding behaviour, oxidative stress status, neurotoxicity, and metabolic requirements).
The introduction provided is well structured focusing on the working hypotheses.
The material and methods are exhaustively documented, and the experimentation is well-designed.
The results are robust, and their interpretation is sound, without any grammar/syntax problems, linguistic or spelling mistakes. The authors finally discuss their findings in respect to the advances in the field, providing a good positioning of their study in the body of current research. Their findings provide the first evidence on the potential adverse effects of microplastics on finfish hatcheries under realistic scenarios (i.e., presence and absence of food) and facilitate further research on this field.
It is a pleasure to have been a reviewer of this study. Due to its excellent quality, I am happy to suggest its publication without any hesitation.
Author Response
We appreciate the reviewer's comments. Thank you.